# K18- and CAG-hACE2 Transgenic Mouse Models and SARS-CoV-2: Implications for Neurodegeneration Research

**DOI:** 10.3390/molecules27134142

**Published:** 2022-06-28

**Authors:** Simona Dedoni, Valeria Avdoshina, Chiara Camoglio, Carlotta Siddi, Walter Fratta, Maria Scherma, Paola Fadda

**Affiliations:** 1Department of Biomedical Sciences, Division of Neuroscience and Clinical Pharmacology, University of Cagliari, 09042 Monserrato, Sardinia, Italy; chiara.camoglio@unica.it (C.C.); c.siddi2@studenti.unica.it (C.S.); wfratta@unica.it (W.F.); pfadda@unica.it (P.F.); 2Department of Neuroscience, Georgetown University Medical Center, Washington, DC 20057, USA; valeria.avdoshina@georgetown.edu; 3Institute of Neuroscience-Cagliari, National Research Council, 00185 Rome, Italy

**Keywords:** COVID-19, K18-transgenic mice, CAG-hACE2 transgenic mice, neuroinvasion, inflammation

## Abstract

COVID-19, caused by severe acute respiratory syndrome coronavirus 2 (SARS-CoV-2), is a global pandemic that might lead to very serious consequences. Notably, mental status change, brain confusion, and smell and taste disorders along with neurological complaints have been reported in patients infected with SARS-CoV-2. Furthermore, human brain tissue autopsies from COVID-19 patients show the presence of SARS-CoV-2 neuroinvasion, which correlates with the manifestation of meningitis, encephalitis, leukocyte infiltration, and neuronal damage. The olfactory mucosa has been suggested as a way of entry into the brain. SARS-CoV-2 infection is also known to provoke a hyper-inflammatory reaction with an exponential increase in the production of pro-inflammatory cytokines leading to systemic responses, even in the absence of direct infection of brain cells. Angiotensin-converting enzyme 2 (ACE2), the entry receptor of SARS-CoV-2, has been extensively demonstrated to be present in the periphery, neurons, and glial cells in different brain regions. To dissect the details of neurological complications and develop therapies helping COVID-19 survivors regain pre-infection quality of life, the development of robust clinical models is highly warranted. Several human angiotensin-converting enzyme 2 (hACE2) transgenic mouse models have been developed and used for antiviral drug screening and vaccine development, as well as for better understanding of the molecular pathogenetic mechanisms of SARS-CoV-2 infection. In this review, we summarize recent results from the studies involving two such mouse models, namely K18- and CAG-hACE2 transgenics, to evaluate the direct and indirect impact of SARS-CoV-2 infection on the central nervous system.

## 1. Introduction

Starting from 2019, the COVID-19 pandemic caused by SARS-CoV-2 hit the world population, and we are currently still dealing with the aftermath of this disease. SARS-CoV-2 appears to act as a “priming” factor triggering the onset of various pathologies affecting the central nervous system (CNS) [1,2,3]. SARS-CoV-2 infection may result in multi-system damage and exacerbate latent neurological disorders. The exact consequences of COVID-19 for the CNS in the long term remain to be described. It is unknown how these people will perform in 5 or 10 years. Young adults have long been thought to deal with COVID-19 quite nicely. So, it remains to be seen how they will perform later in life. Will their young age and plasticity help them compensate well and forget about it, or will COVID-19 have delayed consequences [4]? Basically, we do not know if this can be modeled in mice, which have a different lifespan than humans. The complex interplay between COVID-19 and the pathogenesis of human neurodegenerative diseases has been investigated in only a handful of studies. Due to the lack of long-term results on survivors of COVID-19, it is imperative to develop diverse and representative animal models that would allow studying the outcomes of COVID-19 over time. In this review, we are focusing on the neurotropism, neuropathology, neuroinflammation, and variations in the levels of biomarkers observed in K18- and CAG-hACE2 transgenic mice, which currently represent the most used mouse animal model to study SARS-CoV-2, develop vaccines, and test antiviral molecules. The aim of this review is to summarize the existing information regarding COVID-19-related neuropathology in K18- and CAG-hACE2 mouse models.

## 2. SARS-CoV-2 and Possible Implications for Human Brain Neurodegeneration

SARS-CoV-2 is a virus that belongs to the *Coronaviridae* family. The neurotropism of this family of viruses has been confirmed by their presence in human brain samples, which demonstrates their ability to invade the CNS from the primary infection site [5,6,7]. In a post-mortem case series, SARS-CoV-2 RNA was detected by qRT-PCR in 39% of cryopreserved frontal lobe tissue samples and 50% of formalin-fixed paraffin-embedded medulla oblongata tissue samples. In total, SARS-CoV-2 was found in the brain tissues of 48% of patients who had at least one sample available. Importantly, not only the viral genetic material but also viral proteins, Spike (S), and/or nucleocapsid (N) were detected in the human CNS [8]. Intriguingly, no evidence of viral presence or replication in the vascular endothelial cells of the blood–brain barrier (BBB) was found in patient samples, unlike in recent in vitro studies [9,10]. SARS-CoV-2 is known to use its S glycoprotein to enter the target cells by binding with its receptor, ACE2. Previous studies on a related coronavirus and SARS-CoV-2 have shown that the ACE2 receptor is present in vascular endothelial and smooth muscle cells, in the periphery as well as in the CNS [11,12,13]. Because of the emergence of neurodegenerative consequences of SARS-CoV-2, it is necessary to better understand its possible consequences at the level of the CNS, and it becomes necessary to study the distribution of ACE2 in the human brain in more detail. Yet, another question of how and whether the virus persists in the CNS is a matter of debate among scientists. Considering that detection of viral RNA and proteins in the CNS does not necessarily mean infection, several pressing questions remain unanswered: is SARS-CoV-2 neuroinvasive? If so, does the virus persist in the CNS and what is the likely mechanism of persistence? Are small-molecule antivirals going to be active against SARS-CoV-2 in the brain? Are there any early interventions to mitigate, prevent, or reverse long-term neuropathological consequences? It is still unclear if SARS-CoV-2 is neuroinvasive. There are several hypotheses about how SARS-CoV-2 enters the CNS. The first hypothesis is based on the observation that COVID-19 causes variable olfactory dysfunction [14,15]. Chun et al. identified 17 cohort studies and six case reports with prevalence of olfactory symptoms ranged from 5% to 98% by using the smell identification test (SIT), sinus imaging, and nasoendoscopy [16]. Moreover, in the study by Moein et al., 98% of hospitalized patients had some degree of olfactory dysfunction when evaluated with psychophysical tests, while only 35% of participants self-reported loss of taste or smell [14].

Moreover, neuronal loss has been observed in transgenic mice expressing the hACE2 receptor after intranasal infection with SARS-CoV-2 [17]. Some authors have proposed that the virus infects the nasal epithelium and olfactory nerves and is retrogradely transported to the brain through the axons of neurons [18], but Kahan et al. failed to detect traces of infection of either the olfactory sensory neurons or the olfactory bulb parenchyma [19]. Another possible route of entry of the virus into the CNS is by crossing the BBB due to increased permeability caused by the elevated levels of pro-inflammatory cytokines present in acute COVID-19 disease or infection of the choroid plexus, as the latter expresses the ACE2 receptor [20]. The virus can also be shed by immune cells that migrate into the CNS through the bloodstream. A set of variable symptoms has been observed so far in the virus-infected population, suggesting the detrimental effects of SARS-CoV-2 on the CNS and the increasing evidence of neurological complications it may cause (Figure 1). It is not yet known whether the neurological symptoms are caused directly by the SARS-CoV-2 infiltration into the brain or indirectly by the secondary immune effects of a cytokine storm. In addition, the overproduction of antibodies and the activation of microglia mediated by systemic hypoxia also play an important role, which also leads to neuronal death. To try to answer all these points, studies are underway to evaluate the neuroinvasive potential of the SARS-CoV-2 virus. Neurological complications occur quite commonly in 37% of cases [21,22,23] and are often associated with the presence of more frequent complications related to the severity of the viral infection and a more pronounced acute innate immune response [24]. A growing body of evidence supports the presence of neurological manifestations in both the central and peripheral nervous system in COVID-19 patients [25,26]. Possible immune cross-reactivity between SARS-CoV-2-specific antibodies and various human self-antigens may influence the disease severity, accelerate the development of autoimmunity in susceptible subgroups, and potentially exacerbate autoimmunity in individuals with pre-existing autoimmune diseases [27,28,29]. So far, the neurological impairments associated with COVID-19 have not been thoroughly investigated, with recent research revealing that SARS-CoV-2 causes neuroinflammation and may have serious long-term consequences.

Among the various symptoms, headache, changes in mental status, ataxia, seizures, and stroke-like episodes are the most common central events. Moreover, post-mortem examination of patients with COVID-19 has revealed atrophy and necrosis of the spleen and lymph nodes along with lymphocytic apoptosis [25]. The death of lymphocytes and the surge of cytokines and chemokines are directly related to the innate immune response indicative of SARS-CoV-2 infection. Interestingly, acute necrotizing encephalopathy (ANE), acute disseminated encephalomyelitis (ADEM), and acute inflammatory demyelinating polyneuropathy Guillain–Barré syndrome (GBS) occur in the absence of viral invasion of the CNS, in the presence of systemic cytokine storm, and disruption of the BBB [30,31,32].

SARS-CoV-2 infection is a strong risk factor for the development of complications and long-term neurological consequences that pose a challenge to patient management and treatment decisions.

The possible impact of SARS-CoV-2 virus on the development or pathogenesis of neurodegenerative disorders remains largely unknown. Neurodegenerative diseases such as Alzheimer’s disease (AD), Parkinson’s disease (PD), multiple sclerosis (MS), Huntington’s disease (HD), prion diseases, and others are characterized by a selective and progressive loss of neurons in the CNS, peripheral nervous system, or both, which leads to speech deficits, memory impairment, and non-cognitive symptoms such as depression [33,34]. When most neurons are damaged and subsequently die, neuronal networks can break down and many regions of the brain begin to shrink [35,36,37]. Three-dimensional human brain organoids have been obtained to study the neuroinvasive capacity of the virus by demonstrating that SARS-CoV-2 preferentially infects neurons, causing tau mislocalization from axons to soma, tau hyperphosphorylation, and ultimately neuronal death [38]. Microglia and astrocytes contribute to neuroinflammatory processes associated with neurodegenerative disorders. Microglia activation is known to be associated with the release of pro-inflammatory cytokines, chemokines, and free radicals. Astrocytes exhibit a homeostatic function in the relation to the redox balance, which is disturbed in neurodegeneration. It is also known that the increase in the production of microglial pro-inflammatory mediators is neurotoxic, which is the basis of the etiology of several neurological diseases such as AD and PD. These diseases have been shown to progressively worsen in individuals with COVID-19 due to increased neuroinflammation and microvascular lesions in the brain [39,40,41]. Recently, the association between coronavirus infection and the onset of MS was highlighted, and SARS-CoV-2 infection has been shown to be associated with MS-like signs and symptoms [42,43].

Moreover, several months after infection, COVID-19 survivors frequently report the symptoms such as cognitive impairment, inattention, disorientation, and decreased voluntary movement in response to command [44,45].

In addition to numerous human studies, it is becoming increasingly necessary to create viable preclinical models to study the consequences associated with SARS-CoV-2 infection. Several animal models have already been used to study severe acute respiratory syndrome (SARS), such as mice, rats, hamsters, ferrets, and non-human primates [46].

Mice are the most used laboratory animals in biomedical research, and they have helped scientists develop diagnostics, therapies, and vaccines for multiple human diseases. One of the main challenges in achieving better understanding of neurodegeneration in patients with COVID-19 is to study the virus in suitable animal models.

In the mouse brain, the ACE2 receptor is expressed peripherally with a greater abundance in the liver, testicles, forebrain, spleen, lungs, kidneys, colon, ileum, in the bladder, ventricle, and in the CNS [47]. In fact, Chen et al. reported that ACE2 is expressed in the choroid plexus of the lateral ventricle, substantia nigra pars reticulata, piriform cortex, and olfactory bulb; instead, the hippocampus has low expression of this receptor [48]. One limitation resulting from a previous study of SARS-CoV in murine models is that the ACE2 receptor has a lower affinity to SARS-CoV Spike [47]. For this reason, in order to better reproduce the infection in patients with SARS-CoV-2, mice expressing the hACE2 have been developed (Figure 2). These animals express the hACE2 in various tissues including the brain tissue. Recently, researchers have decided to use the potential of these transgenics to study the infection caused by COVID-19. In this review, we will focus on the consequences that this virus spreads to K18- and CAG-hACE2 transgenic mice.

## 3. SARS-CoV-2-Induced Increased Mortality Rate of K18-hACE2 Mice

In the K-18 transgenic mouse model, originally developed to study the infection of SARS-CoV, h-ACE2 expression is driven by a cytokeratine promoter (K18) in multiple tissues. Dong et al. used RT-PCR to measure hACE2 RNA expression levels in different tissues of both K18-hACE2 mice and humans. High hACE2 expression was detected in the lung, brain, trachea, small intestine, stomach, kidney, large intestine, and testis of the K18-hACE2 mice. In humans, hACE2 expression was found in the lung, stomach, intestine, kidney, and testis. The expression of hACE2 in the brain, trachea, and lung tissues makes the K18-hACE2 transgenic mice model suitable for studying SARS-CoV-2 infection in vivo. However, this study underlined that the K18 promoter leads to a higher hACE2 expression in the brain of K18-hACE2 transgenic mice compared to ACE2 expression in the human brain [49]. This finding underlines the usefulness of K18-hACE2 transgenic mice to study CNS-associated pathologies, even though they do not fully recapitulate human biology.

Several studies have highlighted high coronavirus-infection-associated mortality in this model. McCray et al. in 2007, demonstrated the susceptibility of K18-hACE2 mice to a lethal SARS-CoV infection [50]. Dong et al. used K18-hACE2 mice to evaluate the effect of low and high infectious doses of SARS-CoV-2 at different time points after intranasal inoculation in the brain, colon, and lungs [50]. All the animals lost weight after infection and up to 90% of animals infected with the higher doses (2 × 10^−3^ and 2 × 10^−4^ PFU) died around day 7, while the animals infected with lower doses (2 × 10^−1^ and 2 × 10^−2^ PFU) died by day 10. Only one or two animals from the low-dose groups recovered and survived to the end of the observation period. Viral copy numbers and RNA levels were high in the lungs of animals from all the experimental groups, even at low doses, which confirms the importance of the lungs in the pathogenesis of SARS-CoV-2. In the brain and intestines, viral copy numbers were higher in the high-dose groups compared to the low-dose groups, suggesting the presence of tissue-specific factors that modulate the viral copy number in a dose-dependent manner. Very high levels of viral RNA have been found in the brains of animals infected with higher doses. In addition, they used IHC to measure the N protein of SARS-CoV-2 in cells positive for the neuronal marker (NeuN) and found a strong overlap of the two signals. Accordingly, in the presence of high expression of hACE2 RNA in the same tissues, a high level of hACE2 protein was found. The expression of hACE2 RNA was higher in the lung and in the brain of K18-hACE2 mice. Accordingly, high levels of hACE2 protein were found in the same tissues. The lungs of infected mice appeared severely affected with epithelial damage, alveolar hemorrhage, and lymphocytic infiltration. Congestion was also found in the brain, and its severity increased with the administration of the infectious dose. Overall, in Dong et al.’s study, pathological manifestations in the lungs and brain in animals were similar to those experienced by severely affected COVID-19 patients, confirming the usefulness of K18-hACE2 transgenic mice as a model for better understanding COVID-19 pathogenesis in humans.

## 4. K18-hACE2 Mice and SARS-CoV-2-Evoked Inflammation

Cytokine storm has been described in humans, and it is thought to contribute to the severe respiratory symptoms typical of COVID-19 [51]. Oladunni et al. [52] confirmed the mortality of K18-hACE2 transgenic mice after intranasal inoculation of SARS-CoV-2; in fact, all the animals used in their study died by 6 DPI (days post infection), and at this same time point the presence of a systemic and local chemokine storm was detected, especially in the lungs and brain, where they also detected the presence of the SARS-CoV-2 N protein and hACE2 expression. In the lungs, both chemokine and cytokine levels decreased by 4 DPI. Notably, levels of chemokines and cytokines in the brain increased at 4 DPI, suggesting delayed spread of the virus into the brain. At 6 DPI, high levels of IFN-α were detected in the brain. High levels of chemokines, including MIP-2/CXCL2, IP-10/CXCL10, and MIP-1α/CCL3, were detected in the brain at 6 DPI. The consequence of these high levels of chemokines is the accumulation of inflammatory cells in the tissues of infected mice. Infection of K18-hACE2 transgenic mice induces a cytokine storm localized mainly in the lungs with elevated levels of TNFα, IL-6, IFN-λ, IFN-γ, IFN-α, and several other interleukins (including IL-4 and IL-10).

Interestingly, IP-10/CXCL10 levels were higher in female mice compared to male mice at 2 DPI in the brain. With regard to brain cytokine levels, some sex differences were observed: male mice had higher levels of Th1 and Th17 cytokines compared to female animals, which had a higher Th2 response at 4 DPI, and sex differences normalized at 6 DPI. These data suggest that a cytokine storm in K18-hACE2 transgenic mice mimics in several important aspects the COVID-19 pathology reported in COVID-19 patients [53].

## 5. SARS-CoV-2 Entry, Distribution, and Neuroinvasion in K18-hACE2 Mice

Regarding the spread of the virus, Oladunni et al. described the detection of SARS-CoV-2 N protein in the lungs, brain, and turbinates [52]. Interestingly, some SARS-CoV-2-positive cells have been found in the choroid plexus, suggesting a possible role of the cerebrospinal fluid in the dissemination of the virus to the CNS. The presence of SARS-CoV-2 N protein has also been found in pyramidal neurons in the prefrontal cortex. Despite the presence of SARS-CoV-2 in the brains of infected mice, the histopathological changes identified in the brain tissue were minimal. Few animals showed multifocal perivascular inflammatory and activation of microglial cells, as well as some signs of inflammation and necrosis. Th2 cytokine response was high in the brain, probably as an attempt of the CNS to counterbalance the strong peripheral Th1 response.

In a study by Kumari et al. [17], it was demonstrated that SARS-CoV-2 neuroinvasion leading to encephalitis is associated with mortality of K18-hACE2 transgenic mice. All infected animals died by 6 DPI, and despite a decrease in virus concentrations in the lungs by 5 DPI, the concentration of infectious virus in the brain remained high up to 6 DPI in all animals. On 5 DPI, the animals began to show neurological symptoms such as tremors, ruffled hair, slow movement, and hunchbacked posture. The levels of viral RNA were strongly elevated, and infectious virus particles were found in the brain of infected animals at this time. This indicates a productive infection within the olfactory system, the point of entry of the virus, is taking place. Using H&E staining, the authors were able to detect signs of neuronal death and perivascular hemorrhage. Infected neurons appeared degenerated, and there was leukocyte infiltration in perivascular space and blood vessels. Furthermore, it is possible that SARS-CoV-2 could exploit the capillary vessels from the olfactory mucosa to spread through the brain, as the virus was also detected in blood vessel endothelium [18]. Signs of cell death were found in the cortex, hippocampus, and cerebellum. It was therefore suggested that the virus could enter the brain after an intranasal infection, cross the cribriform plate and infect the neurons close to the point of entry, and then infect other neurons via axonal transportation [17]. No virus was detected in the serum of infected mice, excluding the possibility that the virus could enter the brain through the disruption of the BBB, which is consistent with previous studies [54,55,56,57]. The onset of the most severe symptoms and death of animals in Kumari et al.’s study coincided with peak levels of the virus in the brain and severe neuroinflammation, suggesting that CNS involvement following SARS-CoV-2 infection may be responsible for the severity of the phenotype observed in the infected mice.

In contrast with the previously discussed studies, Zhang et al. confirmed in their work the ability of SARS-CoV-2 to cross the BBB via the transcellular route through infection of brain microvascular endothelial cells (BEMCs) and without disrupting the integrity of tight junctions [58]. Other evidence from mice and humans supports the theory that the virus can enter the CNS through the BBB [59,60]. In their study, Zhang et al. found SARS-CoV-2 in the serum of 67% of infected animals. The S protein has been found in several brain areas, including the cortex, hypothalamus, pons, and medulla oblongata [58]. Contrarily to the previously published studies [61], no virus was detected in the microglial cells and astrocytes despite their strong activation in the brains of infected animals. Using in vitro and in vivo techniques, Zhang et al. demonstrated the ability of the virus to infect BEMCs and cause a strong immune response, which contributes to brain damage and increases vascular permeability, facilitating the penetration of the virus into the CNS. Rothan et al. [62] used primary neuronal cultures derived from K18-hACE2 transgenic mice to further assess the neuronal tropism of SARS-CoV-2 and demonstrated that these cells are permissive for SARS-CoV-2 infection and support active virus replication. In vitro and in vivo upregulation of genes associated with innate immunity and inflammation (including TNF-α, IFN-α, IL-1β, and IL-6) and the activation of pathways involved in necroptosis and neuronal death (ZBP1/pMLKL) were observed.

In a recent paper by Carossino et al. [63], severe symptoms and death of K18-hACE2 transgenic mice were correlated with high viral titers in the brain, hypothermia, and neuroinvasion rather than with the respiratory symptoms commonly seen in humans. Only low levels of hACE2 RNA in neurons were detected, and no hACE2 protein was found. Instead, high expression was found in the olfactory neuroepithelium, suggesting this was the main route for neuroinvasion, but the subsequent neuro-dissemination likely used the hACE2-independent pathway. Other protein factors such as neurophilin-1 have been proposed as a possible receptor responsible for SARS-CoV-2 neuroinvasion but not neuro-dissemination [64], with the exact mechanism in K18-hACE2 transgenic mice remaining poorly understood. Various brain areas, such as the cerebral cortex, hippocampus, hypothalamus, midbrain, pons, medulla oblongata, and brainstem, were found to be severely damaged in the final stage of infection with diffused neurodegeneration, gliosis, and necrosis. Activation of microglial cells and astrocytes worsened the neuronal damage. Severe hypothermia found in severely affected mice correlates with hypothalamic dysfunction due to SARS-CoV-2 infection. Due to high mortality rates and the rapid and severe onset of neuronal complications, K18-hACE2 transgenic mice may not be particularly suitable for evaluating certain treatments [65,66] but serve as one of the most used models to study the neurological complications experienced by COVID-19 patients, including anosmia [67]. Interestingly, the Omicron variant of SARS-CoV-2 lacks the ability to generate neuroinvasion and elicits only mild pulmonary symptoms in K18-hACE2 transgenic mice [68,69]. Recently, Ku et al. [70] successfully used a K18-hACE2 transgenic model variant with enhanced brain susceptibility to SARS-CoV-2 to test the efficacy of a non-integrative lentiviral vector encoding SARS-CoV-2 S protein as a new vaccine. This newly generated B6.K18-hACE2IP-THV transgenic model shows higher expression of hACE2 in the brain; therefore, it is extremely useful not only for studying the treatments and vaccines but also for better understanding the neurological manifestations associated with COVID-19 in humans.

An interesting, recently published study points to the intranasal route of infection in transgenic mice as being responsible for the occurrence of neurological complications in K18-hACE2 transgenic mice. In their work, Fumagalli et al. [71] demonstrated that transgenic mice infected with SARS-CoV-2 via aerosol show strong viral replication in the lungs and induce classic respiratory symptoms and anosmia without leading to fatal neuroinvasion. However, pathology and inflammation of the lungs in aerosol-infected mice lead to more severe consequences compared with intranasal inoculation.

## 6. CAG-hACE2 Transgenic Mice

Tseng et al. [72] was the first group to obtain hACE2 transgenic mouse strains for studying SARS-CoV pathogenesis. In these mice, hACE2 is expressed downstream of a chimeric CAG promoter, comprising the immediate-early enhancer of cytomegalovirus (CMV-IE fused with the chicken β-actin promoter, and the intron from rabbit beta-globin gene). Several lines of transgenic mice were produced, including AC63 and A70, with AC70 displaying strong hACE2 expression in the brain and the lungs. Upon intranasal infection with 10^3^ or 2 × 10^5^ TCID_50_ of the virus, AC70 mice were found to be more susceptible to SARS-CoV infection, and, among the examined tissues, the lungs and brain were the main sites of viral replication [73]. High viral titers were also found in the brains of the same infected animals. Therefore, via the intraperitoneal route, the virus can reach the brain regardless of the route of infection.

In their research, Asaka et al. show through Western blotting analysis that the hACE2 protein is markedly expressed in the brain of CAG-hACE2 mice. Subsequent immunohistochemistry analysis and RT-qPCR confirmed that the N protein levels in the brain increase over time after infection. In addition, the inflammatory pattern at the level of the CNS (levels of mRNA of several cytokines and inflammatory chemokines) shows a peak of expression on day 7 after infection. Based on these results, the infection in the brain was suggested to be secondary to the lung, where a marked upregulation of the inflammatory mediators was observed already 2 days after infection. For these reasons, CAG-hACE2 mice are highly susceptible to SARS-CoV-2, with the lungs and the brain being the main sites of infection [74].

## 7. CAG-hACE2 Transgenic Mice: Research on Antiviral Drugs and Vaccines

Recently, EK1, a pan-coronavirus fusion inhibitor targeting the HR1 domain of S, was developed [75]. Later, the same group performed the X-ray crystallographic analysis of six-helix bundle coreformed by the HR1 and HR2 domains of the SARS-CoV-2 protein subunit S2 and generated a series of lipopeptides derived from EK1 such as EK1C4 [76]. They decided to test the in vivo efficacy of these two peptides on transgenic mice C57BL/6-Tgtn (CAG-human ACE2-IRES-Luciferase-WPRE-polyA) Smoc. Nasal administration of the EK1 peptide resulted in a significant reduction in viral titer in the lung tissues of mice, and EK1C4 at a lower dosage (10 μg/mouse) effectively protected against SARS-CoV-2 infection in their mouse model [77]. The purpose of this study was therefore to demonstrate the efficacy of EK1 and EK1C4 peptides in vivo against several viral variants, forming the foundation of prospective development of human therapies to be used in clinical trials. SARS-CoV-2 S protein is known to be highly glycosylated, which is required for interaction with hACE2 and is believed to help mask the viral epitopes [78,79]. Based on this evidence, Huang et al. immunized CAG-hACE2 transgenic mice using a modified S protein, *N*-glycans trimmed to the mono-GlcNAc-decorated state (S_MG_), missing a glycan shield, to assess its efficacy as a vaccine candidate, and then challenged these animals with some variants of SARS-CoV-2 [80]. The results show that the S_MG_ protein confers effective protection against both the Alpha and Gamma variants.

In addition to the immunodominant receptor-binding domain (RBD), a susceptibility site mapping to the S1 subunit of the S protein is the N-terminal domain (NTD) [81]. Yu et al. [82] have formulated “ReCOV”, a trimeric NTD and RBD two-component SARS-CoV-2 subunit vaccine adjuvanted with BFA03. This recombinant vaccine has been tested in several animal models including CAG-hACE2 mice. Their study was based on the immunization with ReCOV of transgenic female mice, followed by the intranasal infection with SARS-CoV-2. These mice were then euthanized, and samples of lung and brain tissue were collected for the quantification of viral titer. The results obtained showed that this recombinant vaccine could protect these animals from infection with SARS-CoV-2, as the viral load found in both the lungs and the brain of these animals was low, indicating that the RECOV vaccine has a protective effect against SARS-CoV-2 infection in the brain of these mice.

## 8. Other hACE2-Expressing Mouse Models

In addition to the models mentioned above, several mouse models have been developed to express the hACE2 receptor under the control of alternative promoters (Table 1). The hepatocyte nuclear factor 3/forkhead homologue 4 (HFH4) promoter of ciliated epithelial cells is also frequently used [83], such as the syn-hACE2 model, in which the open reading frame of the human ACE2 gene is under the control of a synapse promoter [84]. CMV-hACE2 is another transgenic strain driven by a composite global promoter consisting of the cytomegalovirus (CMV) immediate-early enhancer and the chicken β-actin promoter [72], while in mAce2-hACE, hACE2 expression is driven by the mouse Ace2 promoter (mAce2) [85]. Finally, several different companies are producing a murine model established using CRISPR/Cas9 knock-in technology featuring endogenous hACE2 expression (hACE2 KI). Recently, a novel SARS-CoV-2-sensitive transgenic mouse strain, hACE2 LoxP-Stop (TgCAGLoxPStopACE2GFP), has also been established by crossing hACE2 (LoxP-Stop) mice with Ubi-Cre/ERT2 mice [86]. The generated offspring expressed hACE2 and GFP, after the induction with tamoxifen, which offers the advantage of spatial and temporal control of hACE2 expression to study the role of different tissues in the development of symptoms and complications of COVID-19. In all these models, tissue and cellular expression of hACE2 varies, leading to different levels of viral replication in different organs and cell types and differences in disease pathogenesis.

## 9. Conclusions

Susceptibility to COVID-19 symptoms strongly depends on the entry of SARS-CoV-2 into host cells. The more viruses enter, the greater the susceptibility to viral pathology. Widening antibody responses, lymphocyte dysregulation, and cytokine storms are critical to understanding SARS-CoV-2 infection. In this regard, COVID-19 animal models, including the aforementioned hACE2 transgenic mice, not only help explore the pathogenesis of COVID-19 but also provide useful tools for evaluating vaccines and therapies. The number of some SARS-CoV-2 variants determines complex situations, some being highly infectious. Current studies on COVID-19 pathogenesis are mainly focused on ACE2 and immunopathology, and SARS-CoV-2 infection has been described in various animal models such as macaque, ferret, mouse, and hamster [87,88,89]. Transgenic models have a major limitation in their use to study long-term effects due to the lethality of SARS-CoV-2 disease. However, animal models such as K18-hACE2 transgenic mice have been used for testing the efficacy of neutralizing antibodies against SARS-CoV-2 and its variants [66,90,91]. Signs of lung inflammation and impaired respiratory function were observed in most strains, which would have been the cause of the high incidence, and neurological signs before death were also observed. One option for future studies would be to reduce the viral load in order to let the animals survive and to be able to study the complications over time.

The K18- and CAG-hACE2 transgenic mouse models discussed in this review provide invaluable resources for studying the mechanisms of coronavirus infection, including the risk of neurodegenerative diseases and beyond.

## Figures and Tables

**Figure 1 molecules-27-04142-f001:**
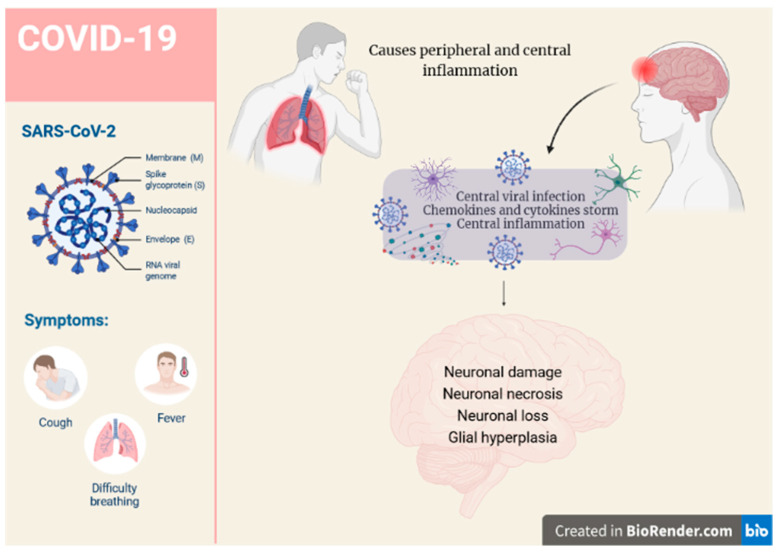
**SARS-CoV-2 in human cells.** SARS-CoV-2 infection activates humoral and cellular immunity, causing lymphocyte dysregulation and cytokine storm, which contribute to the development of peripheral and central damage.

**Figure 2 molecules-27-04142-f002:**
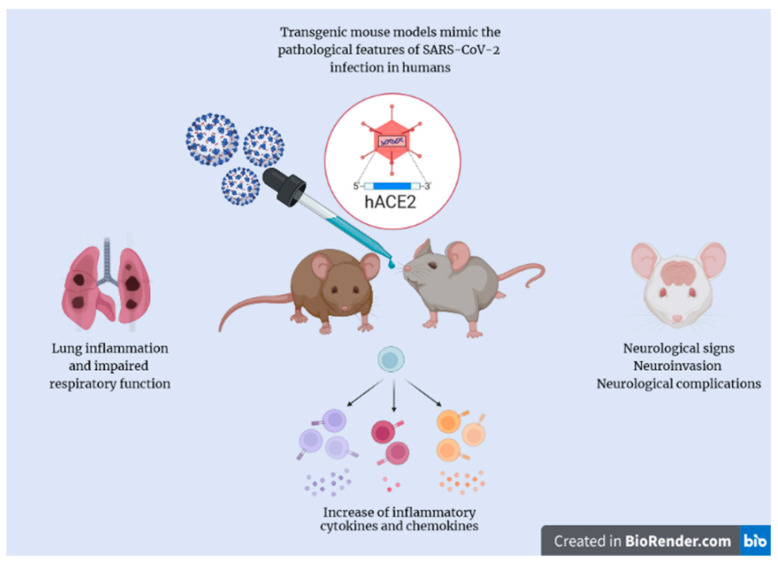
**hACE2 transgenic mouse.** Expression of hACE2 in the mouse offer the possibility of examining different aspects of SARS-CoV-2 infection.

**Table 1 molecules-27-04142-t001:** Promoter regulators in different hACE2 mouse models.

Model	Promoter
K18-hACE	Cytokeratine
CAG-hACE	Cytomegalovirus, β-actin, beta-rabbit globine (CAG)
HFH4-hACE	HFH4 /FOXJ
syn-hACE2	Synapse
CMV-hACE2	Chicken β-actin
mAce2-hACE	Mouse Ace2
hACE2 KI	CRISPR/Cas9 technology

## Data Availability

Data are not contained in the article.

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
