# Peer review of "K18- and CAG-hACE2 Transgenic Mouse Models and SARS-CoV-2: Implications for Neurodegeneration Research"

_molecules, 2022, doi:10.3390/molecules27134142_

Round 1
Reviewer 1 Report
This interesting paper summarizes the current state of research on SARS-CoV-2, and discusses possible routes of infection, interaction with pre-existing neurodegenerative diseases, and therapeutic options.
There are only minor suggestions to improve the narrative balance of different observations mainly on how the virus may enter the brain (see below).
1. line 15: consider to replace “loss of taste” by “smell and taste disorders” (or similar, but including olfaction)
2. line 81: The notion that “It is widely accepted that the virus infects the nasal epithelium and olfactory nerves and is retrogradely transported to the brain through the axons of neurons” is not widely accepted, at least as far it concerns the retrograde neurotransmission through the olfactory nerve. The olfactory route via axons of the ORN or ensheathing glia is controversial. Until now, there are no studies that have documented any implication of the olfactory bulb, in contrast to other brain regions (as mentioned later in this review). Please consider to include relevant literature that contradicts this notion, e.g., Kirschenbaum et al. (Lancet . 2020 Jul 18;396(10245):166. doi: 10.1016/S0140-6736(20)31525-7) or Khan et al. 2021 (Cell 184, 5932–5949; doi.org/10.1016/j.cell.2021.10.027).
3. line 141: What does “… AD´ and PD´ COVID-19…” mean? Do you mean: COVID-19 triggers AD and PD? Please clarify.
4. line 161: replace “pyriform” with “piriform”
5. line 170: Figure2: there is a switch in the description: Replace “neurological sings” with neurological signs”
6. line 173: “SARS-CoV-2-induced mortality of K18-hACE2 mice”. Consider to replace “mortality” by “increased mortality rate” (also further below) or the heading “Increased mortality rate induced by SARS-CoV-2”
7. line 219: insert “of the” between “presence” and “SARS-CoV-2”
8. line 251ff: “The levels of viral RNA were strongly elevated, and infectious virus particles were found in the brain of infected animals at this time. This indicates a productive infection within the olfactory system, the point of entry of the virus, is taking place.” Please clarify that also blood vessel (capillary) routes from the olfactory mucosa are possible transduction ways. Later, line 257ff or line 293ff, the authors describe correctly that various brain areas were involved, however, never the olfactory bulb. This makes a direct olfactory route unlikely (see above).
Author Response
Thanks for the comments and suggestions.
1. line 15: consider replacing “loss of taste” by “smell and taste disorders” (or similar, but including olfaction)
Response 1: “loss of taste” has been modified as suggested.
2. line 81: The notion that “It is widely accepted that the virus infects the nasal epithelium and olfactory nerves and is retrogradely transported to the brain through the axons of neurons” is not widely accepted, at least as far it concerns the retrograde neurotransmission through the olfactory nerve. The olfactory route via axons of the ORN or ensheathing glia is controversial. Until now, there are no studies that have documented any implication of the olfactory bulb, in contrast to other brain regions (as mentioned later in this review). Please consider to include relevant literature that contradicts this notion, e.g., Kirschenbaum et al. (Lancet . 2020 Jul 18;396(10245):166. doi: 10.1016/S0140-6736(20)31525-7) or Khan et al. 2021 (Cell 184, 5932–5949; doi.org/10.1016/j.cell.2021.10.027).
Response 2: We agree with the Reviewer. We have rephrased the sentence: “Some authors have proposed that the virus infects the nasal epithelium and olfactory nerves and is retrogradely transported to the brain through the axons of neurons (18) but Kahan et al. failed to detect traces of infection of either the olfactory sensory neurons or the olfactory bulb parenchyma (19)”
3. line 141: What does “… AD´ and PD´ COVID-19…” mean? Do you mean: COVID-19 triggers AD and PD? Please clarify.
Response 3: We agree with the Reviewer. We have rephrased the sentence:
“It is also known that the increase in the production of microglial pro-inflammatory medi-ators is neurotoxic, which is the basis of the etiology of several neurological diseases such as AD's and PD's. These diseases have been shown to progressively worsen in in-dividuals with COVID-19 due to increased neuroinflammation and microvascular lesions in the brain (39, 40, 41).”
4. line 161: replace “pyriform” with “piriform”
Response 4: Done
5. line 170: Figure2: there is a switch in the description: Replace “neurological sings” with neurological signs”
Response 5: Done
6. line 173: “SARS-CoV-2-induced mortality of K18-hACE2 mice”. Consider to replace “mortality” by “increased mortality rate” (also further below) or the heading “Increased mortality rate induced by SARS-CoV-2”
Response 6: Done
7. line 219: insert “of the” between “presence” and “SARS-CoV-2”
Response 7: Done
8. line 251ff: “The levels of viral RNA were strongly elevated, and infectious virus particles were found in the brain of infected animals at this time. This indicates a productive infection within the olfactory system, the point of entry of the virus, is taking place.” Please clarify that also blood vessel (capillary) routes from the olfactory mucosa are possible transduction ways. Later, line 257ff or line 293ff, the authors describe correctly that various brain areas were involved, however, never the olfactory bulb. This makes a direct olfactory route unlikely (see above).
Response 8: The sentence “Furthermore, it is possible that SARS-CoV-2 could exploit the capillary vessels from the olfactory mucosa to spread through the brain, as the virus was also detected in blood vessel endothelium (18)” was added to clarify the point suggested by the referee.
Reviewer 2 Report
In this review Dedoni et al summarise existing information regarding Covid-19 -related neuropathology in K18- and CAG-hACE2 murine models. Their main focus of attention was directed to neurotropism, neuropathology, neuroinflammation, and variations in the levels of biomarkers observed in K18- and CAG-hACE2 transgenic mice. The authors state that these 2 mouse lines are currently the most used murine animal models when looking at SARS-CoV-2, vaccine development and antiviral molecule testing. The possible impact of SARS-CoV-2 virus on the development or pathogenesis of neurodegenerative disorders remains largely unknown and SARS-CoV-2 infection has been shown to be associated with MS-like signs and symptoms.
The authors state that several months after infection, COVID-19 survivors frequently report symptoms such as cognitive impairment, inattention, disorientation, and decreased voluntary movement in response to command. There is a real demand for suitable animal models to allow a better understanding of neurodegeneration in patients with Covid-19.
The authors evaluate the role of transgenic mouse models and conclude that they provide invaluable resources for studying the mechanisms of coronavirus infection, including the risk of neurodegenerative diseases. The K18-hACE2 transgenic mice are a very important resource for testing the efficacy of neutralising antibodies against SARS-CoV-2 and its variants. The CAG-hACE2 transgenic mice are able to address different aspects of the SARS-CoV-2 disease pathogenesis.
Overall this is a well written, nicely presented review that covers some extremely relevant references and discusses the fine detail and the differences found in those publications.
Minor Comments:
- It is really interesting that the authors state that patient samples had no evidence of viral presence or replication in the vascular endothelial cells of the blood brain barrier, while recent in vitro studies found the opposite.
- The authors have highlighted many useful, as yet unanswered questions throughout this review article, and have assessed a number of studies that may shed some light on a number of issues.
- The authors state that Covid-19 causes variable olfactory dysfunction with ranges between 5 and 98% which I find incredible. Can the authors please comment on this?
- The authors cover the main areas of interest in this review and assess topics in a reasonably robust manner. They highlight the importance of the information that is still to be uncovered and analysed and they illuminate the reasons for using the mouse models discussed in this manuscript.
- The authors make a very important point that in addition to various human studies, there is very definite requirement for the creation of viable preclinical animal models to allow the study of the consequences that are associated with a SARS-CoV-2 infection.
- The supporting Figures in this review are helpful and informative and the text is easy to read.
- The authors review the state of the literature confirming the utility of K18-hACE2 mice as a model for better understanding Covid-19 pathogenesis in humans, although they may not be that suitable for evaluating certain treatments. The CAG-hACE2 mice provide a different platform for SARS-CoV-2 studies by demonstrating that the lungs and brain are the main sites of infection. Transgenic mouse models do have major limitations with regards to their use in long-term studies basically due to the lethality of a SARS-CoV-2 infection.
- The references that have been utilised and discussed seem appropriate for this particular review.
- There are a few basic typing errors here and there (such as lines 69 and 70) with the odd word missing and a few grammatical errors (such as line 270, 381-382, 383). These just need a bit of tidying up.
Author Response
We thank the Reviewer for the positive comments.
Minor Comments:
- It is really interesting that the authors state that patient samples had no evidence of viral presence or replication in the vascular endothelial cells of the blood brain barrier, while recent in vitro studies found the opposite.
Response 1: From Shimmel et al., 2021 – “Human endothelial cells express very low levels of the SARS-CoV-2 receptor ACE2 and the protease TMPRSS2, which blocks their capacity for productive viral infection, and limits their capacity to produce infectious virus. Accordingly, endothelial cells can only be infected when they overexpress ACE2 or are exposed to very high concentrations of SARS-CoV-2. These data suggest that in vivo, endothelial cells are unlikely to be infected with SARS-CoV-2 and that infection may only occur if the adjacent pulmonary epithelium is denuded (basolateral infection) or a high viral load is present in the blood (apical infection)”. It is possible that in vitro studies allow higher viral concentrations to reach the cells, making them more vulnerable to a productive infection.
2. The authors have highlighted many useful, as yet unanswered questions throughout this review article, and have assessed a number of studies that may shed some light on a number of issues.
3. The authors state that Covid-19 causes variable olfactory dysfunction with ranges between 5 and 98% which I find incredible. Can the authors please comment on this?
Response 3: The comment from Chung et al, was added in the text to clarify the variable range between 5 and 98%. “Chun et al. identified 17 cohort studies and 6 case reports with prevalence of olfactory symptoms ranged from 5% to 98% by using the smell identification test (SIT), sinus imaging, and nasoendoscopy (16). Moreover, in the study by Moein et al., 98% of hospitalized patients had some degree of olfactory dysfunction when evaluated with psychophysical tests, while only 35% of participants self-reported loss of taste or smell (14).”
The literature on the variability of olfactory dysfunctions from SARS-CoV-2 is very heterogeneous also due to studies that are based on subjective measurements, such as self-reported questionnaires, without evaluating the presence of hyposmia with validated tests. Furthermore, there is often no information on the temporal course of hyposmia and its correlation between clinical and viral recovery.
4. The authors cover the main areas of interest in this review and assess topics in a reasonably robust manner. They highlight the importance of the information that is still to be uncovered and analysed and they illuminate the reasons for using the mouse models discussed in this manuscript.
5. The authors make a very important point that in addition to various human studies, there is very definite requirement for the creation of viable preclinical animal models to allow the study of the consequences that are associated with a SARS-CoV-2 infection.
6. The supporting Figures in this review are helpful and informative and the text is easy to read.
7. The authors review the state of the literature confirming the utility of K18-hACE2 mice as a model for better understanding Covid-19 pathogenesis in humans, although they may not be that suitable for evaluating certain treatments. The CAG-hACE2 mice provide a different platform for SARS-CoV-2 studies by demonstrating that the lungs and brain are the main sites of infection. Transgenic mouse models do have major limitations with regards to their use in long-term studies basically due to the lethality of a SARS-CoV-2 infection.
8. The references that have been utilised and discussed seem appropriate for this particular review.
9. There are a few basic typing errors here and there (such as lines 69 and 70) with the odd word missing and a few grammatical errors (such as line 270, 381-382, 383). These just need a bit of tidying up.
Response 9: Typos were corrected